# Nonnegative Tensor Completion
# via Integer Optimization

**Caleb Xavier Bugg**[†]        **Chen Chen**[‡]        **Anil Aswani**[†]

[†]University of California, Berkeley, {caleb_bugg,aaswani}@berkeley.edu
[‡]The Ohio State University, chen.8018@osu.edu

## Abstract

Unlike matrix completion, tensor completion does not have an algorithm that is known to achieve the information-theoretic sample complexity rate. This paper develops a new algorithm for the special case of completion for nonnegative tensors. We prove that our algorithm converges in a linear (in numerical tolerance) number of oracle steps, while achieving the information-theoretic rate. Our approach is to define a new norm for nonnegative tensors using the gauge of a particular 0-1 polytope; integer linear programming can, in turn, be used to solve linear separation problems over this polytope. We combine this insight with a variant of the Frank-Wolfe algorithm to construct our numerical algorithm, and we demonstrate its effectiveness and scalability through computational experiments using a laptop on tensors with up to one-hundred million entries.

## 1 Introduction

Tensors generalize matrices. A tensor $\psi$ of *order* $p$ is $\psi \in \mathbb{R}^{r_1 \times \cdots \times r_p}$, where $r_i$ is the *dimension* of the tensor in the $i$-th index, for $i = 1, \ldots, p$. Though related, many problems that are polynomial-time solvable for matrices are NP-hard for tensors. For instance, it is NP-hard to compute the rank of a tensor (Hillar and Lim, 2013). Tensor versions of the spectral norm, nuclear norm, and singular value decomposition are also NP-hard to compute (Hillar and Lim, 2013; Friedland and Lim, 2014).

### 1.1 Past Approaches to Tensor Completion

Tensor completion is the problem of observing (possibly with noise) a subset of entries of a tensor and then estimating the remaining entries based on an assumption of low-rankness. The tensor completion problem is encountered in a number of important applications, including computer vision (Liu et al., 2012; Zhang et al., 2019), regression with only categorical variables (Aswani, 2016), healthcare (Gandy et al., 2011; Dauwels et al., 2011), and many other application domains (Song et al., 2019).

Although the special case of matrix completion is now well understood, the tensor version of this problem has an unresolved tension. To date, no tensor completion algorithm has been shown to achieve the information-theoretic sample complexity rate. Namely, for a tensor completion problem on a rank $k$ tensor with sample size $n$, the information-theoretic rate for estimation error is $\sqrt{k \cdot \sum_i r_i / n}$ (Gandy et al., 2011). In fact, evidence suggests a computational barrier in which no polynomial-time algorithm can achieve this rate (Barak and Moitra, 2016). One set of approaches has polynomial-time computation but requires exponentially more samples than the information-theoretic rate (Gandy et al., 2011; Mu et al., 2014; Barak and Moitra, 2016; Montanari and Sun, 2018), whereas another set of approaches achieve substantially closer to the information-theoretic rate but (in order to attain global minima) requires solving NP-hard problems that lack numerical algorithms (Chandrasekaran et al., 2012; Yuan and Zhang, 2016, 2017; Rauhut and Stojanac, 2021).

36th Conference on Neural Information Processing Systems (NeurIPS 2022).

However, algorithms that achieve the information-theoretic rate have been developed for a few special cases of tensors. Completion of nonnegative rank-1 tensors can be written as a convex optimization problem (Aswani, 2016). For symmetric orthogonal tensors, a variant of the Frank-Wolfe algorithm has been proposed (Rao et al., 2015). Though this latter paper does not prove their algorithm achieves the information-theoretic rate, this can be shown using standard techniques (Gandy et al., 2011). This latter paper is closely related to the approach we take in this paper, with one of the key differences being the design of a different separation oracle in order to support a different class of tensors.

## 1.2 Contributions

This paper proposes a numerical algorithm for completion of nonnegative tensors that provably converges to a global minimum in a linear (in numerical tolerance) number of oracle steps, while achieving the information-theoretic rate. Nonnegative tensors are encountered in many applications. For instance, image and video data usually consists of nonnegative tensors (Liu et al., 2012; Zhang et al., 2019). Notably, the image demosaicing problem that must be solved by nearly every digital camera (Li et al., 2008) is an instance of a nonnegative tensor completion problem, though it has not previously been interpreted as such. Nonnegative tensor completion is also encountered in specific instances of recommender systems (Song et al., 2019), healthcare applications (Gandy et al., 2011; Dauwels et al., 2011), and statistical regression contexts (Aswani, 2016).

Our approach is to define a new norm for nonnegative tensors using the gauge of a specific 0-1 polytope that we construct. We prove this norm acts as a convex surrogate for rank and has low statistical complexity (as measured by the Rademacher average for the tensor, viewed as a function from a set of indices to the corresponding entry), but is NP-hard to approximate to an arbitrary accuracy. Importantly, we prove that the tensor completion problem using this norm achieves the information-theoretic rate in terms of sample complexity. However, the resulting tensor completion problem is NP-hard to solve. Nonetheless, because our new norm is defined using a 0-1 polytope, we can use integer linear optimization as a practical means to solve linear separation problems over the polytope. We combine this insight with a variant of the Frank-Wolfe algorithm to construct our numerical algorithm, and we demonstrate its effectiveness and scalability through experiments.

## 2 Norm for Nonnegative Tensors

Consider a tensor $\psi \in \mathbb{R}^{r_1 \times \cdots \times r_p}$. To refer to a specific entry in the tensor we use the notation $\psi_x := \psi_{x_1, \ldots, x_p}$, where $x = (x_1, \ldots, x_p)$, and $x_i \in [r_i]$ denotes the value of the $i$-th index, with $[s] := \{1, \ldots, s\}$. Let $\rho = \sum_i r_i$, $\pi = \prod_i r_i$, and $\mathcal{R} = [r_1] \times \cdots \times [r_p]$.

A nonnegative rank-1 tensor is $\psi_x = \prod_{k=1}^{p} \theta_{x_k}^{(k)}$, where $\theta^{(k)} \in \mathbb{R}_+^{r_k}$ are nonnegative vectors indexed by the different values of $x_k \in [r_k]$. To simplify notation, we drop the superscript in $\theta_{x_k}^{(k)}$ and write this as $\theta_{x_k}$ when clear from the context. For a nonnegative tensor $\psi$, its nonnegative rank is

$$\mathrm{rank}_+(\psi) = \min\{q \mid \psi = \textstyle\sum_{k=1}^{q} \psi^k, \psi^k \in \mathcal{B}_\infty \text{ for } k \in [q]\}, \tag{1}$$

where we define the ball of nonnegative rank-1 tensors whose maximum entry is $\lambda \in \mathbb{R}_+$ to be

$$\mathcal{B}_\lambda = \{\psi : \psi_x = \lambda \cdot \textstyle\prod_{k=1}^{p} \theta_{x_k}, \ \theta_{x_k} \in [0, 1], \text{ for } x \in \mathcal{R}\} \tag{2}$$

and $\mathcal{B}_\infty = \lim_{\lambda \to \infty} B_\lambda$. A nonnegative CP decomposition is given by the summation $\psi = \sum_{k=1}^{\mathrm{rank}_+(\psi)} \psi^k$, where $\psi^k \in \mathcal{B}_\infty$ for $k \in [\mathrm{rank}_+(\psi)]$.

### 2.1 Convex Hull of Nonnegative Rank-1 Tensors

Now for $\lambda \in \mathbb{R}_+$ consider the finite set of points:

$$\mathcal{S}_\lambda = \{\psi : \psi_x = \lambda \cdot \textstyle\prod_{k=1}^{p} \theta_{x_k}, \ \theta_{x_k} \in \{0, 1\} \text{ for } x \in \mathcal{R}\}. \tag{3}$$

Using standard techniques from integer optimization (Hansen, 1979; Padberg, 1989), the above set of points can be rewritten as a set of linear constraints on binary variables: $\mathcal{S}_\lambda = \{\psi : \lambda \cdot (1 - p) + \lambda \cdot \sum_{k=1}^{p} \theta_{x_k} \leq \psi_x, 0 \leq \psi_x \leq \lambda \cdot \theta_{x_k}, \theta_{x_k} \in \{0, 1\}, \text{ for } k \in [p], x \in \mathcal{R}\}$. Our first result is that the convex hulls of the set of points $\mathcal{S}_\lambda$ and of the ball $\mathcal{B}_\lambda$ are equivalent.

**Proposition 2.1.** *We have the relation that* $\mathcal{C}_\lambda := \mathrm{conv}(\mathcal{B}_\lambda) = \mathrm{conv}(\mathcal{S}_\lambda)$.

*Proof.* We prove this by showing the two set inclusions $\operatorname{conv}(\mathcal{S}_\lambda) \subseteq \operatorname{conv}(\mathcal{B}_\lambda)$ and $\operatorname{conv}(\mathcal{B}_\lambda) \subseteq \operatorname{conv}(\mathcal{S}_\lambda)$. The first inclusion is immediate since by definition we have $\mathcal{S}_\lambda \subset \mathcal{B}_\lambda$, and so we focus on proving the second inclusion. We prove this by contradiction: Suppose $\operatorname{conv}(B_\lambda) \not\subset \operatorname{conv}(\mathcal{S}_\lambda)$. Then there exists a tensor $\psi' \in B_\lambda$ with $\psi' \notin \operatorname{conv}(\mathcal{S}_\lambda)$. By the hyperplane separation theorem, there exists $\varphi \in \mathbb{R}^{r_1 \times \cdots \times r_p}$ and $\delta > 0$ such that $\langle \varphi, \psi' \rangle \geq \langle \varphi, \psi \rangle + \delta$ for all $\psi \in \operatorname{conv}(\mathcal{S}_\lambda)$, where $\langle \cdot, \cdot \rangle$ is the usual inner product that is defined as the summation of elementwise multiplication. Now consider the multilinear optimization problem

$$
\begin{aligned}
\max \ & \langle \varphi, \psi \rangle \\
\text{s.t. } & \psi_x = \lambda \cdot \prod_{k=1}^{p} \theta_{x_k}, \quad \text{for } x \in \mathcal{R} \\
& \theta_{x_k} \in [0,1], \qquad\quad \text{for } x \in \mathcal{R}
\end{aligned}
\tag{4}
$$

Proposition 2.1 of (Drenick, 1992) shows there exists a global optimum $\psi''$ of (4) with $\psi'' \in \mathcal{S}_\lambda$. By construction, we must have $\langle \varphi, \psi'' \rangle \geq \langle \varphi, \psi' \rangle$, which implies $\langle \varphi, \psi'' \rangle \geq \langle \varphi, \psi \rangle + \delta$ for all $\psi \in \operatorname{conv}(\mathcal{S}_\lambda)$. But this last statement is a contradiction since $\psi'' \in \mathcal{S}_\lambda \subseteq \operatorname{conv}(\mathcal{S}_\lambda)$. $\square$

*Remark* 2.2. This result has two useful implications. First, $\mathcal{C}_\lambda$ is a polytope since it is the convex hull of a finite number of bounded points. Second, the elements of $\mathcal{S}_\lambda$ are the vertices of $\mathcal{C}_\lambda$, since any individual element cannot be written as a convex combination of the other elements.

We will call the set $\mathcal{C}_\lambda$ the nonnegative tensor polytope. A useful observation is that the following relationships hold: $\mathcal{B}_\lambda = \lambda \mathcal{B}_1$, $\mathcal{S}_\lambda = \lambda \mathcal{S}_1$, and $\mathcal{C}_\lambda = \lambda \mathcal{C}_1$.

## 2.2 Constructing a Norm for Nonnegative Tensors

Since the set of nonnegative tensors forms a cone (Qi et al., 2014), we need to use a modified definition of a norm. A norm on a cone $\mathcal{K}$ is a function $p : \mathcal{K} \to \mathbb{R}_+$ such that for all $x, y \in \mathcal{K}$ the function has the following three properties: $p(x) = 0$ if and only if $x = 0$; $p(\gamma \cdot x) = \gamma \cdot p(x)$ for $\gamma \in \mathbb{R}_+$; and $p(x + y) \leq p(x) + p(y)$. The difference with the usual norm definition is subtle: The second property here is required to hold for $\gamma \in \mathbb{R}_+$, whereas in the usual norm definition we require $p(\gamma \cdot x) = |\gamma| \cdot p(x)$ for all $\gamma \in \mathbb{R}$.

We next use $\mathcal{C}_\lambda$ to construct a new norm for nonnegative tensors using a gauge (or Minkowski functional) construction. Though constructing norms using a gauge is common in machine learning (Chandrasekaran et al., 2012), the convex sets used in these constructions are symmetric about the origin. Symmetry guarantees that scaling the ball eventually includes the entire space (Rockafellar and Wets, 2009). However, in our case $\mathcal{C}_\lambda$ is not symmetric about the origin, and so without proof we do not *a priori* know whether scaling $\mathcal{C}_1$ eventually includes the entire space of nonnegative tensors. Thus we have to explicitly prove the gauge is a norm.

**Proposition 2.3.** *The function defined as*

$$
\|\psi\|_+ := \inf\{\lambda \geq 0 \mid \psi \in \lambda \mathcal{C}_1\}
\tag{5}
$$

*is a norm for nonnegative tensors $\psi \in \mathbb{R}_+^{r_1 \times \cdots \times r_p}$.*

*Proof.* We first prove the above function is finite. Consider any nonnegative tensor $\psi \in \mathbb{R}_+^{r_1 \times \cdots \times r_p}$, and note there exists a decomposition $\psi = \sum_{i=1}^{\pi} \psi^k$ with $\psi^k \in \|\psi\|_{\max} \cdot \mathcal{B}_1$ (Qi et al., 2016). (This holds because we can choose each $\psi^k$ to have a single non-zero value that corresponds to a different entry of $\psi$.) Hence by Proposition 2.1, $\psi^k \in \|\psi\|_{\max} \cdot \mathcal{C}_1$. Recalling the decomposition of $\psi$, this means $\psi \in \pi \|\psi\|_{\max} \cdot \mathcal{C}_1$ which by definition means $\|\psi\|_+ \leq \pi \|\psi\|_{\max}$. Thus $\|\psi\|_+$ must be finite.

Next we verify that the three properties of a norm on a cone are satisfied. To do so, we first observe that by definition: $\mathcal{C}_1$ is convex; $0 \in \mathcal{C}_1$; and $\mathcal{C}_1$ is closed and bounded. Thus by Example 3.50 of (Rockafellar and Wets, 2009) we have $\{\psi : \|\psi\|_+ = 0\} = \{0\}$, which means the first norm property holds. Also by Example 3.50 of (Rockafellar and Wets, 2009) we have $\lambda \mathcal{C}_1 \subseteq \lambda' \mathcal{C}_1$ for all $0 \leq \lambda \leq \lambda'$, which means the second norm property holds. Last, note Example 3.50 of (Rockafellar and Wets, 2009) establishes sublinearity of the gauge, and so the third norm property holds. $\square$

## 2.3 Convex Surrogate for Nonnegative Tensor Rank

An important property of our norm $\|\psi\|_+$ is that it can be used as a convex surrogate for tensor rank, whereas the max and Frobenius norms cannot.

**Proposition 2.4.** *If $\psi$ is a nonnegative tensor, then we have $\|\psi\|_{\max} \leq \|\psi\|_+ \leq \mathrm{rank}_+(\psi) \cdot \|\psi\|_{\max}$. If $\mathrm{rank}_+(\psi) = 1$, then $\|\psi\|_+ = \|\psi\|_{\max}$.*

*Proof.* Consider any $\lambda \geq 0$. If $\psi \in \mathcal{S}_\lambda$, then by definition $\|\psi\|_{\max} = \lambda$. By the convexity of norms we have that: if $\psi \in \mathcal{C}_\lambda$, then $\|\psi\|_{\max} \leq \lambda$. This means that for all $\lambda \geq 0$ we have $\mathcal{C}_\lambda \subseteq \mathcal{U}_\lambda := \{\psi : \|\psi\|_{\max} \leq \lambda\}$. Thus we have the relation $\inf\{\lambda \geq 0 \mid \psi \in \lambda\mathcal{U}_1\} \leq \inf\{\lambda \geq 0 \mid \psi \in \lambda\mathcal{C}_1\}$. But the left side is $\|\psi\|_{\max}$ and the right side is $\|\psi\|_+$. This proves the left side of the inequality.

Next consider the case where $\mathrm{rank}_+(\psi) = 1$. By definition we have $\psi \in \|\psi\|_{\max} \cdot \mathcal{B}_1$. Thus $\psi \in \|\psi\|_{\max} \cdot \mathcal{C}_1$, and so $\|\psi\|_+ \leq \|\psi\|_{\max}$. This proves the rank-1 case.

Last we prove the right side of the desired inequality. Consider any nonnegative tensor $\psi$. Recall its nonnegative CP decomposition $\psi = \sum_{k=1}^{\mathrm{rank}_+(\psi)} \psi^k$ with $\psi^k \in \mathcal{B}_\infty$. The triangle inequality gives

$$\|\psi\|_+ \leq \sum_{k=1}^{\mathrm{rank}_+(\psi)} \|\psi^k\|_+ = \sum_{k=1}^{\mathrm{rank}_+(\psi)} \|\psi^k\|_{\max}, \tag{6}$$

where the equality follows from the above-proved rank-1 case since $\mathrm{rank}_+(\psi^k) = 1$ by definition of the CP decomposition. But $\|\psi^k\|_{\max} \leq \|\psi\|_{\max}$ since the tensors are nonnegative. $\square$

*Remark* 2.5. These bounds are tight. The lower and upper bounds are achieved by all nonnegative rank-1 tensors. The identity matrix with $k$ columns achieves the upper bound with $\mathrm{rank}_+(\psi) = k$.

# 3 Measures of Norm Complexity

## 3.1 Computational Complexity

We next characterize the computational complexity of our new norm on nonnegative tensors.

**Proposition 3.1.** *It is NP-hard to approximate the nonnegative tensor norm $\|\cdot\|_+$ to arbitrary accuracy.*

*Proof.* The dual norm is $\|\varphi\|_\circ = \sup\{\langle\varphi, \psi\rangle \mid \|\psi\|_+ \leq 1\}$ for all tensors $\varphi \in \mathbb{R}^{r_1 \times \cdots \times r_p}$. Theorems 3 and 10 of (Friedland and Lim, 2016) show that approximation of the dual norm $\|\cdot\|_\circ$ is polynomial-time reducible to approximation of the norm $\|\cdot\|_+$. We proceed by showing a polynomial-time reduction to approximation of $\|\cdot\|_\circ$. Without loss of generality assume $p = 2$ and $d := r_1 = r_2$. The appendix of (Witsenhausen, 1986) proves that MAX CUT is polynomial-time reducible to $\sup\{\langle\varphi, \psi\rangle \mid \psi \in \mathcal{S}_1\}$. However, since the objective function is linear and the set $\mathcal{S}_1$ consists of the vertices of $\mathcal{C}_1$, this means that optimization problem is equivalent to $\sup\{\langle\varphi, \psi\rangle \mid \psi \in \mathcal{C}_1\}$. This is the desired polynomial-time reduction of MAX CUT to approximation of the dual norm $\|\cdot\|_\circ$ because $\mathcal{C}_1 = \{\psi : \|\psi\|_+ \leq 1\}$. The result now follows by recalling that approximately solving MAX CUT to arbitrary accuracy is NP-hard (Papadimitriou and Yannakakis, 1991; Arora et al., 1998). $\square$

**Corollary 3.2.** *Given $K \in \mathbb{R}_+$ and $\psi \in \mathbb{R}_+^{r_1 \times \cdots \times r_p}$, it is NP-complete to determine if $\|\psi\|_+ \leq K$.*

*Proof.* The problem is NP-hard by reduction from the problem of Proposition 3.1. In particular, a binary search can approximate the norm using this decision problem, $(\|\psi\|_+ \leq K)$?, as an oracle. By Proposition 2.4, the search can be initialized over the interval $[0, \pi\|\psi\|_{\max}]$. Furthermore, the decision problem is in NP because we can use the (polynomial-sized) $\theta$ from (2) as a certificate. $\square$

## 3.2 Stochastic Complexity of Norm

We next show that our norm $\|\psi\|_+$ has low stochastic complexity. Let $X = \{x\langle 1\rangle, \ldots, x\langle n\rangle\}$, and suppose $\sigma_i$ are independent and identically distributed (i.i.d.) Rademacher random variables (i.e., $\sigma_i = \pm 1$ with probability $\frac{1}{2}$) (Bartlett and Mendelson, 2002; Srebro et al., 2010). The Rademacher complexity for a set of functions $\mathcal{H}$ is $R(\mathcal{H}) = \mathbb{E}(\sup_{h \in \mathcal{H}} \frac{1}{n}|\sum_{i=1}^n \sigma_i \cdot h(x\langle i\rangle)|)$, and the *worst case* Rademacher complexity of $\mathcal{H}$ is $W(\mathcal{H}) = \sup_X \mathbb{E}_\sigma(\sup_{h \in \mathcal{H}} \frac{1}{n}|\sum_{i=1}^n \sigma_i \cdot h(x\langle i\rangle)|)$. These notions can be used to measure the complexity of sets of matrices (Srebro and Shraibman, 2005) or tensors (Aswani, 2016). The idea is to interpret each nonnegative tensor as a function $\psi : \mathcal{R} \to \mathbb{R}_+$ from a set of indices $x \in \mathcal{R}$ to the corresponding entry of the tensor $\psi_x$. This complexity notion is useful for the completion problem because it can be directly translated into generalization bounds.

**Proposition 3.3.** *We have* $\mathsf{R}(\mathcal{C}_\lambda) \leq \mathsf{W}(\mathcal{C}_\lambda) \leq 2\lambda\sqrt{\rho/n}$.

*Proof.* First note that from their definitions we get $\mathsf{R}(\mathcal{C}_\lambda) \leq \mathsf{W}(\mathcal{C}_\lambda)$. Next define the set $\mathcal{P}_\lambda = \{\pm\psi : \psi \in \mathcal{S}_\lambda\}$, and recall that $\mathcal{C}_\lambda = \mathrm{conv}(\mathcal{S}_\lambda)$ by Proposition 2.1. This means $\mathsf{W}(\mathcal{C}_\lambda) = \mathsf{W}(\mathcal{S}_\lambda)$ (Ledoux and Talagrand, 1991; Bartlett and Mendelson, 2002). Next observe that

$$\begin{aligned}
\mathsf{W}(\mathcal{C}_\lambda) = \mathsf{W}(\mathcal{S}_\lambda) &= \sup_X \mathbb{E}_\sigma \big( \sup_{\psi \in \mathcal{S}_\lambda} \tfrac{1}{n} \big| \textstyle\sum_{i=1}^n \sigma_i \cdot \psi_{x\langle i\rangle} \big| \big) \\
&= \sup_X \mathbb{E}_\sigma \big( \max_{\psi \in \mathcal{P}_\lambda} \tfrac{1}{n} \cdot \textstyle\sum_{i=1}^n \sigma_i \cdot \psi_{x\langle i\rangle} \big) \leq \sup_X r\sqrt{2\log \#\mathcal{P}_\lambda}/n
\end{aligned} \tag{7}$$

where in the second line we replaced the supremum with a maximum, since the set $\mathcal{P}_\lambda$ is finite, and used the Finite Class Lemma (Massart, 2000) with $r = \max_{\psi \in \mathcal{P}_\lambda} \sqrt{\sum_{i=1}^n (\psi_{x\langle i\rangle})^2} \leq \lambda\sqrt{n}$. This inequality on $r$ is due to the fact that $\mathcal{P}_\lambda$ consists of tensors whose entries are from $\{-\lambda, 0, \lambda\}$. Thus $\mathsf{W}(\mathcal{C}_\lambda) \leq \lambda\sqrt{(2\log 2)\cdot(\rho+1)/n}$. The result follows by noting that $\log 2 \cdot (\rho+1) \leq 2\rho$. $\square$

*Remark* 3.4. Because $\psi$ has $\pi = O(r^p)$ entries, the Rademacher complexity in a typical tensor norm (e.g., max and Frobenius norms) will be $O(\sqrt{\pi/n}) = O(\sqrt{r^p/n})$. However, the Rademacher complexity in our norm $\|\psi\|_+$ is $O(\sqrt{\rho/n}) = O(\sqrt{rp/n})$, which is exponentially smaller.

# 4   Tensor Completion

Suppose we have data $(x\langle i\rangle, y\langle i\rangle) \in \mathcal{R} \times \mathbb{R}$ for $i = 1, \ldots, n$. Let $I = \{i_1, \ldots, i_u\} \subseteq [n]$ be any set of points that specify all the unique $x\langle i\rangle$ for $i = 1, \ldots, n$, meaning the set $U = \{x\langle i_1\rangle, \ldots, x\langle i_u\rangle\}$ does not have any repeated points and $x\langle i\rangle \in U$ for all $i = 1, \ldots, n$. The nonnegative tensor completion problem using our norm $\|\psi\|_+$ is given by

$$\begin{aligned}
\widehat{\psi} \in \arg\min_\psi \; &\tfrac{1}{n} \textstyle\sum_{i=1}^n \big( y\langle i\rangle - \psi_{x\langle i\rangle} \big)^2 \\
\text{s.t. } &\|\psi\|_+ \leq \lambda
\end{aligned} \tag{8}$$

We will study the statistical properties of the above estimate and describe the elements of algorithm to solve the above optimization problem. The purpose of defining the set $U$ is that it is used in constructing the algorithm used to solve the above problem.

## 4.1   Statistical Guarantees

Though in the previous section we calculated a Rademacher complexity for nonnegative tensors in $\mathcal{C}_\lambda$ viewed as functions, here we use an alternative approach to derive generalization bounds. The reason is that generalization bounds using the Rademacher complexity are not tight here.

Our approach is based on the observation that the nonnegative tensor completion problem (8) using our norm $\|\psi\|_+$ is equivalent to a *convex aggregation* problem (Nemirovski, 2000; Tsybakov, 2003; Lecué et al., 2013) for a finite set of functions. In particular, by Proposition 2.1 we have $\{\psi : \|\psi\|_+ \leq \lambda\} = \mathcal{C}_\lambda = \mathrm{conv}(\mathcal{S}_\lambda)$. The implication is we can directly apply existing results for convex aggregation to provide a tight generalization bound for the solution of (8).

**Proposition 4.1** ((Lecué et al., 2013))**.** *Suppose* $|y| \leq b$ *almost surely. Given any* $\delta > 0$*, with probability at least* $1 - 4\delta$ *we have that*

$$\mathbb{E}\big((y - \widehat{\psi}_x)^2\big) \leq \min_{\varphi \in \mathcal{C}_\lambda} \mathbb{E}\big((y - \varphi_x)^2\big) + c_0 \cdot \max\big[b^2, \lambda^2\big] \cdot \max\big[\zeta_n, \tfrac{\log(1/\delta)}{n}\big], \tag{9}$$

*where* $c_0$ *is an absolute constant and*

$$\zeta_n = \begin{cases} \dfrac{2^\rho}{n}, & \text{if } 2^\rho \leq \sqrt{n} \\ \sqrt{\dfrac{1}{n} \log\Big(\dfrac{e2^\rho}{\sqrt{n}}\Big)}, & \text{if } 2^\rho > \sqrt{n} \end{cases} \tag{10}$$

*Remark* 4.2. We make two comments. First, note that $\zeta_n = O(\sqrt{\rho/n})$. Second, in some regimes $\zeta_n$ can be considerably faster than the $\sqrt{\rho/n}$ rate.

Generalization bounds under specific noise models, such as an additive noise model, follow as a corollary to the above proposition combined with Proposition 2.4.

**Corollary 4.3.** *Suppose $\varphi$ is a nonnegative tensor with $\mathrm{rank}_+(\varphi) = k$ and $\|\varphi\|_{\max} \leq \mu$. If $(x\langle i \rangle, y\langle i \rangle)$ are independent and identically distributed with $|y\langle i \rangle - \varphi_{x\langle i \rangle}| \leq e$ almost surely and $\mathbb{E}y\langle i \rangle = \varphi_{x\langle i \rangle}$. Then given any $\delta > 0$, with probability at least $1 - 4\delta$ we have*

$$\mathbb{E}\big((y - \widehat{\psi}_x)^2\big) \leq e^2 + c_0 \cdot \big(\mu k + e\big)^2 \cdot \max\big[\zeta_n, \tfrac{\log(1/\delta)}{n}\big], \tag{11}$$

*where $\zeta_n$ is as in (10) and $c_0$ is an absolute constant.*

*Remark* 4.4. The above result achieves the information-theoretic rate when the rank $k = O(1)$.

## 4.2 Computational Complexity

Though (8) is a convex optimization problem, our next result shows that solving it is NP-hard.

**Proposition 4.5.** *It is NP-hard to solve the optimization problem (8) to an arbitrary accuracy.*

*Proof.* Define the ball of radius $\delta > 0$ centered at a nonnegative tensor $\psi$ to be $B(\psi, \delta) = \{\varphi : \|\varphi - \psi\|_F \leq \delta\}$. Next define $W(\mathcal{C}_1, \delta) = \bigcup_{\psi \in \mathcal{C}_1} B(\psi, \delta)$ and $W(\mathcal{C}_1, -\delta) = \{\psi \in \mathcal{C}_1 : B(\psi, \delta) \subseteq \mathcal{C}_1\}$. The weak membership problem for $\mathcal{C}_1$ is that given a nonnegative tensor $\psi$ and a $\delta > 0$ decide whether $\psi \in W(\mathcal{C}_1, \delta)$ or $\psi \notin W(\mathcal{C}_1, -\delta)$. Theorem 10 of (Friedland and Lim, 2016) shows that approximation of the norm $\|\cdot\|_+$ is polynomial-time reducible to the weak membership problem for $\mathcal{C}_1$. Since Proposition 3.1 shows that approximation of the norm $\|\cdot\|_+$ is NP-hard, the result follows if we can reduce the weak membership problem to (8).

Suppose we are given inputs $\psi$ and $\delta$ for the weak membership problem. Choose $x\langle i \rangle$ for $i = 1, \ldots, \pi$ such that each element in $\mathcal{R}$ is enumerated exactly once. Next choose $y\langle i \rangle = \psi_{x\langle i \rangle}$. Finally, note if we solve (8) and the minimum objective value is less than or equal to $\delta$, then we have $\psi \in W(\mathcal{C}_1, \delta)$; otherwise we have $\psi \notin W(\mathcal{C}_1, -\delta)$. The result now follows since this was the desired reduction. $\square$

We note that the decision version of (8), that is ascertaining whether a given tensor $\psi$ attains an objective less than a given value while also satisfying $\|\psi\|_+ \leq \lambda$, is NP-complete. This follows directly from Corollary 3.2 and Proposition 4.5.

## 4.3 Numerical Computation

Although it is NP-hard, there is substantial structure that enables practical numerical computation of global minima of (8). The key observation is that $\mathcal{C}_1$ is a 0-1 polytope, which implies the linear separation problem on this polytope can be solved using integer linear optimization. Integer optimization has well-established global algorithms that are guaranteed to solve the separation problem for this polytope. This is a critical feature that enables the use of the Frank-Wolfe algorithm or one of its variants to solve (8) to a desired numerical tolerance. In fact, the Frank-Wolfe variant known as *Blended Conditional Gradients* (BCG) (Braun et al., 2019) is particularly well-suited for calculating a solution to (8) for the following reasons:

The first reason is that our problem has structure such that the BCG algorithm will terminate in a linear (with respect to numerical tolerance) number of oracle steps. A sufficient condition for this linear convergence is if the feasible set is a polytope and the objective function is strictly convex over the feasible set. For our problem (8), the objective function can be made strictly convex over the feasible set by an equivalent reformulation. Specifically, we use the equivalent reformulation in which we change the feasible set from $\{\psi : \|\psi\|_+ \leq \lambda\} = C_\lambda$ to $\mathrm{Proj}_U(\mathcal{C}_\lambda)$ where the projection is done over the unique indices specified by the set $U$. In fact, this projection is trivial to implement because it simply consists of discarding the entries of $\psi$ that are not observed by the $x\langle i \rangle$ indices.

The second is that the BCG algorithm uses weak linear separation, which accommodates early-termination of the associated integer linear optimization. Integer optimization software tends to find optimal or near-optimal solutions considerably faster than certifying the optimality of such solutions. Furthermore, a variety of tuning parameters (Berthold et al., 2018) can be used to accelerate the search for good primal solutions when exact solutions are not needed. We also deploy a fast alternating maximization heuristic in order to avoid integer optimization oracle calls where possible. Thus, early-termination allows us to deploy a globally convergent algorithm with practical solution times.

Hence we use the BCG algorithm to compute global minima of (8) to arbitrary numerical tolerance. For brevity we omit a full description of the algorithm, and instead focus on the separation oracle,

---
**Algorithm 1:** Weak Separation Oracle for $\mathcal{C}_\lambda$
---
**Input:** linear objective $c \in \mathbb{R}^{r_1 \times \cdots \times r_p}$, point $\psi \in \mathcal{C}_\lambda$, accuracy $K \geq 1$, gap estimate $\Phi > 0$, norm bound $\lambda$

**Output:** Either (1) vertex $\varphi \in \mathcal{S}_\lambda$ with $\langle c, \psi - \varphi \rangle \geq \Phi/K$, or (2) **false:** $\langle c, \psi - \varphi \rangle \leq \Phi$ for all $\varphi \in \mathcal{C}_\lambda$
---

---
**Algorithm 2:** Alternating Maximization
---
**Input:** linear objective $c \in \mathbb{R}^{r_1 \times \cdots \times r_p}$, point $\psi \in \mathcal{C}_\lambda$, norm bound $\lambda$, incumbent (binary) solution $\hat{\theta} \in \mathcal{S}_\lambda$

**Output:** Best known solution $\theta$

$\theta \leftarrow \hat{\theta}$
$z \leftarrow z_M(\theta)$
**for** $i = 1$ **to** $p$ **do**
   **for** $k = 1$ **to** $r_i$ **do**
      $\theta_k^{(i)} \leftarrow 1 - \theta_k^{(i)}$
      **if** $z_M(\theta) > z$ **then**
         $z \leftarrow z_M(\theta)$
      **else**
         $\theta_k^{(i)} \leftarrow 1 - \theta_k^{(i)}$
      **end if**
   **end for**
**end for**
---

which is the main component specifically adapted to our application. The oracle is described in Algorithm 1, where $\langle \cdot, \cdot \rangle$ is the dot product for tensors viewed as vectors; for notational simplicity we state the oracle in terms of the original space, ignoring projection onto $U$. Output condition (1) provides separation with some vertex $\varphi$, while (2) requires certification that no such vertex exists.

In implementing the weak separation oracle, we first attempt separation with our alternating maximization procedure. Described as Algorithm 2, it involves the following objective function:

$$z_M(\theta) := \sum_{x \in \mathcal{R}} \langle c_x, \psi_x - \lambda \cdot \prod_{k=1}^p \theta_{x_k} \rangle \tag{12}$$

The separation problem is thus treated as a multilinear binary optimization problem, and the algorithm successively minimizes in each dimension. We apply this heuristic a fixed number of times, randomly complementing entries in the incumbent each time. The procedure runs in polynomial-time, and is not guaranteed to separate nor can it certify that such separation is impossible. However, in our experiments it succeeds often, offering substantial practical speedups.

If alternating maximization fails to separate, then we solve the integer programming problem:

$$
\begin{aligned}
\max_{\varphi, \theta} \ &\langle c, \psi - \varphi \rangle \\
\text{s.t. } &\lambda \cdot (1 - p) + \lambda \cdot \sum_{k=1}^p \theta_{x_k} \leq \varphi_x && x \in \mathcal{R} \\
&0 \leq \varphi_x \leq \lambda \cdot \theta_{x_k} && k \in [p], x \in \mathcal{R} \\
&\theta_{x_k} \in \{0, 1\} && k \in [p], x \in \mathcal{R}
\end{aligned}
\tag{13}
$$

Note that early termination is deployed: we stop whenever an incumbent solution is found such that the objective is greater than $\Phi/K$. If no such solution exists, then the integer programming solver is guaranteed to (eventually) establish a dual bound $z^U$ such that $\langle c, \psi - \varphi \rangle \leq z^U \leq \Phi$.

## 5 Numerical Experiments

Here we present results that show the efficacy and scalability of our algorithm for nonnegative tensor completion. Our experiments were conducted on a laptop computer with 8GB of RAM and an Intel Core i5 2.3Ghz processor with 2-cores/4-threads. The algorithms were coded in Python 3. We used

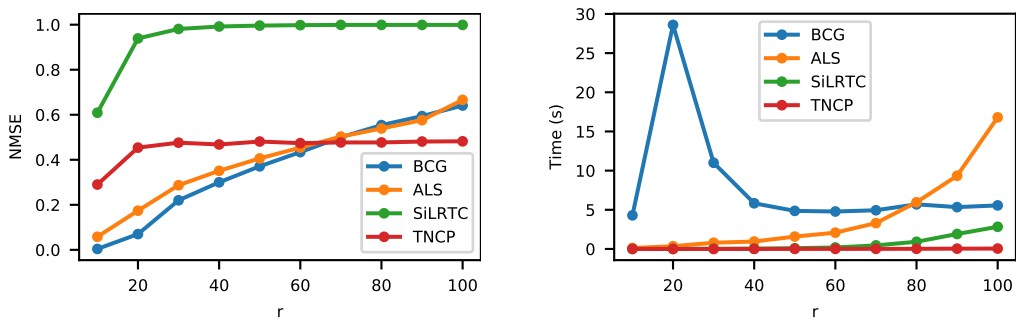

Figure 1: Results for order-3 nonnegative tensors with size $r \times r \times r$ and $n = 500$ samples.

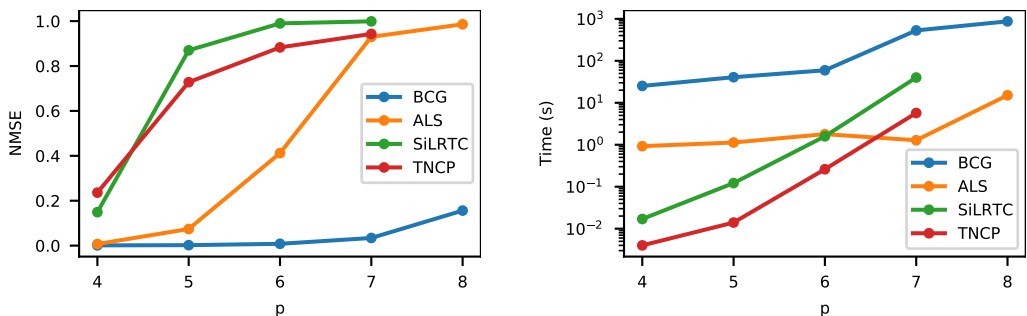

Figure 2: Results for increasing order nonnegative tensors with size $10^{\times p}$ and $n = 10,000$ samples.

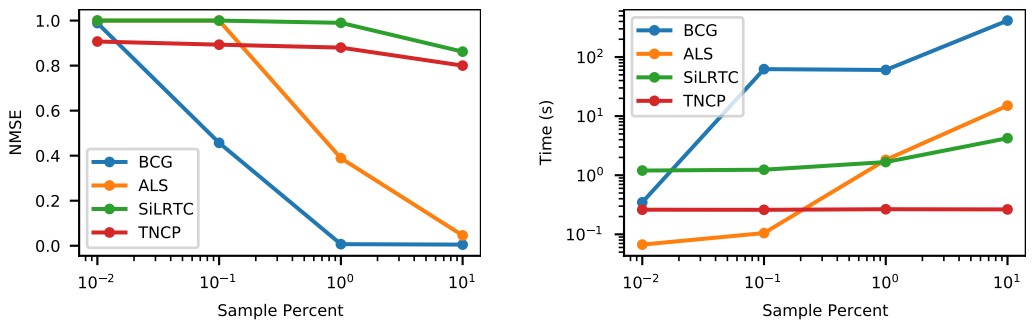

Figure 3: Results for nonnegative tensors with size $10^{\times 6}$ and increasing $n$ samples.

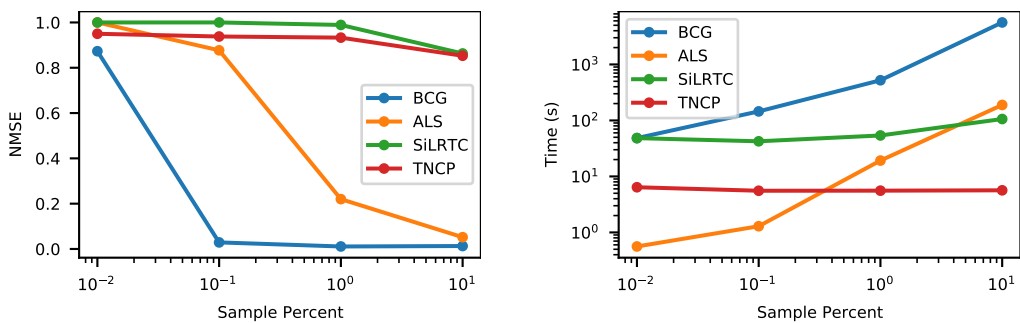

Figure 4: Results for nonnegative tensors with size $10^{\times 7}$ and increasing $n$ samples.

Gurobi v9.1 (Gurobi Optimization, LLC, 2021) to solve the integer programs (13). As a benchmark, we use alternating least squares (ALS) – which is often called a "workhorse" for numerical tensor problems (Kolda and Bader, 2009) – as well as two state-of-the-art methods implemented by the PyTen package (Song et al., 2019), namely the simple low rank tensor completion (SiLRTC) algorithm (Liu et al., 2012) and the trace norm regularized CP decomposition (TNCP) algorithm (Liu et al., 2014). PyTen is available from `https://github.com/datamllab/pyten` under a GPL 2 license.

To minimize the impact of hyperparameter selection in our numerical results, we provided the ground truth values when possible. For instance, in our nonnegative tensor completion formulation (8) we chose $\lambda$ to be the smallest value for which we could certify that $\|\psi\|_+ \leq \lambda$ for the true tensor $\psi$. This was accomplished by construction of the true tensor $\psi$. For ALS and TNCP, we used a nonnegative rank $k$ that was the smallest value for which we could certify that $\text{rank}_+(\psi) \leq k$. This was also accomplished by construction of the true tensor $\psi$. A last note is that ALS often works better when used with L2 regularization (Navasca et al., 2008). The hyperparameter for the L2 regularization for ALS was chosen in a way favorable to ALS in order to maximize its accuracy.

## 5.1 Experiments with Third-Order Tensors

Our first set of results concerns tensors of order $p = 3$ with increasing dimensions. In each experiment, the true tensor $\psi$ was constructed by randomly choosing 10 points from $\mathcal{S}_1$ and then taking a random convex combination. This construction ensures $\|\psi\|_+ \leq 1$ and $\text{rank}_+(\psi) \leq 10$. We used $n = 500$ samples (with indices sampled with replacement). Each experiment was repeated 100 times, and the results are shown in Figure 1. A table of these values and their standard error is found in the Appendix. We measured accuracy using normalized mean squared error (NMSE) $\|\widehat{\psi} - \psi\|_F^2 / \|\psi\|_F^2$, which is a more stringent measure than used in Corollary 4.3 because the statistical theoretical result does not include normalization. The results show that our approach provides modestly higher accuracy but requires more computation time. However, computation time remains on the order of seconds for the various tensor sizes. We note that the NMSE value of approximately $0.48$ that TNCP converges to is the value achieved by a tensor that is identically the average of $y\langle i \rangle$ in all its entries.

## 5.2 Experiments with Increasing Tensor Order

Our second set of results concerns tensors with increasing order $p$, where each dimension takes the value $r_i = 10$ for $i = 1, \ldots, p$. In each experiment, the true tensor was constructed by the method in Sec. 5.1. We used $n = 10,000$ samples (with indices sampled with replacement). Each experiment was repeated 100 times, and the results are shown in Figure 2 with the full values and standard error given in the Appendix. SiLRTC and TNCP were unable to run for tensors with $10^8$ entries. The results show that our approach provides much higher accuracy but requires more computation time. However, computation remains on the order of minutes even for the largest tensor with $10^8$ entries.

## 5.3 Experiments with Increasing Sample Size

Our third and fourth set of results concerns tensors of size $10^{\times 6}$ and $10^{\times 7}$, respectively, where experiments are conducted with increasing sample size, given in the table as percentage of total entries. The true tensor $\psi$ was constructed as in Sec. 5.1. We begin with sample percentage 0.01% (with indices sampled with replacement), and extend each set of experiments' sample size by one order of magnitude. Each experiment was repeated 100 times, and the results are shown in Figures 3 and 4 with the full values and standard error given in the Appendix. Again, the numerical results show that our approach provides substantially higher accuracy but requires more computation time. The computation time remains on the order of minutes for most of the sampling schemes, excepting the draw of $10^6$ entries for a tensor with $10^7$ entries (i.e., about 1.5 hours).

## 6 Conclusion

We defined a new norm for nonnegative tensors and used it to develop an algorithm for nonnegative tensor completion that provably converges in a linear number of oracle steps and meets the information-theoretic sample complexity rate. Its efficacy and scalability were demonstrated using experiments. The next step is to generalize this approach to all tensors. In fact, our norm definitions, optimization formulations, algorithm design, and theoretical analysis all extend to general tensors.

**Acknowledgements**

This material is based upon work partially supported by the NSF under grant CMMI-1847666.

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
