# Appendix: Tables of Numerical Experiment Results

Table 1: Results for order-3 nonnegative tensors with $n = 500$ samples

| Tensor Size | BCG with Weak Oracle | | Alternating Least Squares | |
| --- | --- | --- | --- | --- |
| | NMSE | Time (s) | NMSE | Time (s) |
| $10 \times 10 \times 10$ | $0.004 \pm 0.001$ | $4.29 \pm 0.194$ | $0.058 \pm 0.002$ | $0.122 \pm 0.003$ |
| $20 \times 20 \times 20$ | $0.070 \pm 0.003$ | $28.6 \pm 0.753$ | $0.174 \pm 0.005$ | $0.351 \pm 0.012$ |
| $30 \times 30 \times 30$ | $0.220 \pm 0.004$ | $11.0 \pm 0.671$ | $0.287 \pm 0.005$ | $0.796 \pm 0.030$ |
| $40 \times 40 \times 40$ | $0.300 \pm 0.004$ | $5.83 \pm 0.202$ | $0.351 \pm 0.005$ | $0.947 \pm 0.043$ |
| $50 \times 50 \times 50$ | $0.371 \pm 0.004$ | $4.86 \pm 0.090$ | $0.406 \pm 0.006$ | $1.58 \pm 0.093$ |
| $60 \times 60 \times 60$ | $0.434 \pm 0.004$ | $4.78 \pm 0.063$ | $0.454 \pm 0.006$ | $2.08 \pm 0.105$ |
| $70 \times 70 \times 70$ | $0.500 \pm 0.004$ | $4.94 \pm 0.063$ | $0.503 \pm 0.008$ | $3.30 \pm 0.140$ |
| $80 \times 80 \times 80$ | $0.554 \pm 0.004$ | $5.73 \pm 0.112$ | $0.539 \pm 0.010$ | $5.96 \pm 0.226$ |
| $90 \times 90 \times 90$ | $0.594 \pm 0.004$ | $5.34 \pm 0.077$ | $0.576 \pm 0.011$ | $9.34 \pm 0.302$ |
| $100 \times 100 \times 100$ | $0.641 \pm 0.004$ | $5.56 \pm 0.072$ | $0.666 \pm 0.013$ | $16.8 \pm 0.568$ |

| Tensor Size | Simple Low Rank TC (SiLRTC) | | Trace Norm & ADMM (TNCP) | |
| --- | --- | --- | --- | --- |
| | NMSE | Time (s) | NMSE | Time (s) |
| $10 \times 10 \times 10$ | $0.609 \pm 0.002$ | $0.004 \pm 0.000$ | $0.290 \pm 0.004$ | $0.003 \pm 0.000$ |
| $20 \times 20 \times 20$ | $0.939 \pm 0.000$ | $0.007 \pm 0.000$ | $0.454 \pm 0.005$ | $0.002 \pm 0.000$ |
| $30 \times 30 \times 30$ | $0.981 \pm 0.000$ | $0.019 \pm 0.000$ | $0.476 \pm 0.004$ | $0.003 \pm 0.000$ |
| $40 \times 40 \times 40$ | $0.992 \pm 0.000$ | $0.043 \pm 0.000$ | $0.468 \pm 0.004$ | $0.004 \pm 0.000$ |
| $50 \times 50 \times 50$ | $0.996 \pm 0.000$ | $0.095 \pm 0.001$ | $0.481 \pm 0.004$ | $0.006 \pm 0.000$ |
| $60 \times 60 \times 60$ | $0.998 \pm 0.000$ | $0.190 \pm 0.002$ | $0.474 \pm 0.004$ | $0.010 \pm 0.000$ |
| $70 \times 70 \times 70$ | $0.999 \pm 0.000$ | $0.456 \pm 0.003$ | $0.477 \pm 0.003$ | $0.014 \pm 0.000$ |
| $80 \times 80 \times 80$ | $0.999 \pm 0.000$ | $0.918 \pm 0.008$ | $0.477 \pm 0.003$ | $0.020 \pm 0.000$ |
| $90 \times 90 \times 90$ | $0.999 \pm 0.000$ | $1.91 \pm 0.032$ | $0.481 \pm 0.003$ | $0.040 \pm 0.001$ |
| $100 \times 100 \times 100$ | $0.999 \pm 0.000$ | $2.83 \pm 0.050$ | $0.482 \pm 0.003$ | $0.050 \pm 0.001$ |

Table 2: Results for increasing order nonnegative tensors and $n = 10,000$ samples

| Tensor Size | BCG with Weak Oracle | | Alternating Least Squares | |
| --- | --- | --- | --- | --- |
| | NMSE | Time (s) | NMSE | Time (s) |
| $10^{\times 4}$ | $0.001 \pm 0.000$ | $25.2 \pm 1.11$ | $0.007 \pm 0.001$ | $0.925 \pm 0.025$ |
| $10^{\times 5}$ | $0.002 \pm 0.000$ | $40.7 \pm 0.82$ | $0.074 \pm 0.006$ | $1.13 \pm 0.038$ |
| $10^{\times 6}$ | $0.008 \pm 0.001$ | $59.5 \pm 1.20$ | $0.412 \pm 0.025$ | $1.79 \pm 0.097$ |
| $10^{\times 7}$ | $0.034 \pm 0.004$ | $530 \pm 368$ | $0.930 \pm 0.019$ | $1.28 \pm 0.050$ |
| $10^{\times 8}$ | $0.156 \pm 0.014$ | $872 \pm 66$ | $0.986 \pm 0.010$ | $15.1 \pm 0.199$ |

| Tensor Size | Simple Low Rank TC (SiLRTC) | | Trace Norm & ADMM (TNCP) | |
| --- | --- | --- | --- | --- |
| | NMSE | Time (s) | NMSE | Time (s) |
| $10^{\times 4}$ | $0.149 \pm 0.004$ | $0.017 \pm 0.001$ | $0.236 \pm 0.002$ | $0.004 \pm 0.000$ |
| $10^{\times 5}$ | $0.870 \pm 0.002$ | $0.122 \pm 0.005$ | $0.728 \pm 0.004$ | $0.014 \pm 0.000$ |
| $10^{\times 6}$ | $0.990 \pm 0.000$ | $1.59 \pm 0.053$ | $0.883 \pm 0.003$ | $0.261 \pm 0.000$ |
| $10^{\times 7}$ | $0.999 \pm 0.000$ | $40 \pm 1.23$ | $0.943 \pm 0.002$ | $5.71 \pm 0.010$ |
| $10^{\times 8}$ | $*$ | $*$ | $*$ | $*$ |

The $*$ symbol indicates that the algorithm was unable to run for the given tensor size.

Table 3: Results for nonnegative tensors with size $10^{\times 6}$ and increasing $n$ samples

| Sample Percent | BCG with Weak Oracle | | Alternating Least Squares | |
| --- | --- | --- | --- | --- |
| | NMSE | Time (s) | NMSE | Time (s) |
| 0.01 | $0.989 \pm 0.002$ | $0.348 \pm 0.016$ | $1.000 \pm 0.000$ | $0.067 \pm 0.001$ |
| 0.1 | $0.457 \pm 0.023$ | $62.4 \pm 4.79$ | $1.000 \pm 0.000$ | $0.105 \pm 0.001$ |
| 1 | $0.007 \pm 0.000$ | $60.4 \pm 1.32$ | $0.389 \pm 0.023$ | $1.82 \pm 0.098$ |
| 10 ($n = 100,000$) | $0.005 \pm 0.000$ | $413.7 \pm 11.1$ | $0.046 \pm 0.005$ | $15.03 \pm 0.401$ |
| Sample Percent | Simple Low Rank TC (SiLRTC) | | Trace Norm & ADMM (TNCP) | |
| | NMSE | Time (s) | NMSE | Time (s) |
| 0.01 | $1.000 \pm 0.000$ | $1.20 \pm 0.055$ | $0.907 \pm 0.004$ | $0.261 \pm 0.001$ |
| 0.1 | $1.000 \pm 0.000$ | $1.24 \pm 0.041$ | $0.893 \pm 0.003$ | $0.260 \pm 0.001$ |
| 1 | $0.990 \pm 0.000$ | $1.67 \pm 0.053$ | $0.880 \pm 0.003$ | $0.266 \pm 0.001$ |
| 10 ($n = 100,000$) | $0.862 \pm 0.001$ | $4.23 \pm 0.141$ | $0.800 \pm 0.003$ | $0.264 \pm 0.001$ |

Table 4: Results for nonnegative tensors with size $10^{\times 7}$ and increasing $n$ samples

| Sample Percent | BCG with Weak Oracle | | Alternating Least Squares | |
| --- | --- | --- | --- | --- |
| | NMSE | Time (s) | NMSE | Time (s) |
| 0.01 | $0.873 \pm 0.016$ | $48.4 \pm 3.42$ | $1.000 \pm 0.000$ | $0.561 \pm 0.003$ |
| 0.1 | $0.029 \pm 0.004$ | $145 \pm 36.6$ | $0.877 \pm 0.023$ | $1.29 \pm 0.043$ |
| 1 | $0.011 \pm 0.001$ | $522 \pm 11.7$ | $0.220 \pm 0.016$ | $19.2 \pm 0.603$ |
| 10 ($n = 1,000,000$) | $0.013 \pm 0.001$ | $5623 \pm 381$ | $0.052 \pm 0.007$ | $189 \pm 4.89$ |
| Sample Percent | Simple Low Rank TC (SiLRTC) | | Trace Norm & ADMM (TNCP) | |
| | NMSE | Time (s) | NMSE | Time (s) |
| 0.01 | $1.000 \pm 0.000$ | $48.3 \pm 7.95$ | $0.950 \pm 0.002$ | $6.41 \pm 0.152$ |
| 0.1 | $1.000 \pm 0.000$ | $42.4 \pm 1.46$ | $0.938 \pm 0.002$ | $5.55 \pm 0.007$ |
| 1 | $0.989 \pm 0.000$ | $53.9 \pm 1.96$ | $0.933 \pm 0.002$ | $5.57 \pm 0.007$ |
| 10 ($n = 1,000,000$) | $0.863 \pm 0.001$ | $106 \pm 3.44$ | $0.853 \pm 0.002$ | $5.65 \pm 0.013$ |