# OpenReview forum: "Nonnegative Tensor Completion via Integer Optimization"
_NeurIPS.cc/2022/Conference — NeurIPS 2022 Accept_

### Official Review · Reviewer_4C1A · 2022-07-01

**Rating:** 6
**Confidence:** 4
**Soundness:** 3 good
**Presentation:** 2 fair
**Contribution:** 3 good

**Summary:**

This paper studies the tensor completion problem for tensors with non-negative entries. The paper proposes a non-negative analog of the nuclear norm, denoted $\Vert \cdot \Vert_{+}$ whose 1-ball is defined to be a convex combination of rank-1 $\{0, 1\}$-tensors. The paper proposes to solve the problem, dual to $\Vert \cdot \Vert_{+}$-minimization to recover unknown non-negative tensor from a subset of observed entries. This approach is a natural analog of the nuclear norm minimization for general tensors. The paper proves that this approach w.h.p. can recover the unknown tensor from an essentially statistically optimal number of unknown samples. The authors prove that this minimization problem is NP-hard. At the same time, they show that the BCG algorithm can solve the problem with a linear number of calls to a linear separation oracle. Moreover, the paper proposes a heuristic algorithm for the oracle and studies its performance on synthetic data.

**Questions:**

1) Can you please clarify what results in the papers cited on lines 33-34 achieve information-theoretic bounds? Are there specific numbered theorems? As far as I know, for $n\times n \times n$ tensors of rank r, Yuan and Zhang'16 prove recovery from r^{1/2}n^{3/2} samples, which is not the information-theoretic rate ($\widetilde{O}(rn)$). I was not able to locate the corresponding information-theoretic rate results in the other two cited papers. I would like to kindly suggest to include exact statements of the prior work that achieves information-theoretic bounds in the appendix, as they are quite rare and are directly related to the main contribution of this paper.

2) Cor 4.3 does not seem to achieve the information-theoretic bound unless the rank $k = O(1)$. Do I understand correctly that if $e = 0$, Cor 4.3 proves recovery from essentially $k^4\cdot \rho$ entries (as opposed to $k\cdot \rho$)? If that's correct, I believe the authors should either explicitly say that $k = O(1)$, or they should be more explicit in the description of their results.

3)  The abstract and the text of the paper contain the claim: "We prove that our algorithm converges in a linear (in numerical tolerance) number of oracle steps, while achieving the information-theoretic rate." However, the nature of the oracle does not seem to be mentioned until the end of the paper. In particular, it is not mentioned that this oracle solves an NP-hard problem. I believe those are important details, which if mentioned early give a better understanding of the results of the paper.

4) Are there any additional simple assumptions that guarantee that the oracle can be implemented in polynomial time? This will give specific assumptions under which your algorithm for (8) works in polynomial time. Note that for standard nuclear norm (NN) minimization there are some regimes in which problem can be solved in polynomial time, even though the general problem is NP-hard. For example, using SOS, NN minimization is known for tensors with orthogonal components by Potechin-Steurer'17 and for low-rank tensors with random components by Kivva-Potechin'20.

5) Prop. 3.1 and Cor 3.3 are a bit hard to understand, and I needed to look into the paper of Lecue et al to understand the claim. For example, in Prop 3.1, what is y? This variable seems to have meaning only when comes in pair with x, i.e., in pair $x\langle i\rangle, y\langle i\rangle$. It will also simplify reading if the definition of $\mathbb{E}$ used in L197 is included. I believe it just stands for average over all entries of the tensor, but there are other ways to interpret it. What is the difference between $\psi$ and $\widehat{\psi}$ in Prop. 4.1 and Cor 4.3?

6) Did you try to run your algorithm on any real-world dataset? How does it compare to state-of-the-art approaches on such datasets? How does your algorithm compare to Liu-Moitra NeurIPS'20?

7) In line 199 it seems that $o(\sqrt{\rho/n})$ should be $O(\sqrt{\rho/n})$. How can $\xi_n$ be faster than $O(\sqrt{\rho/n})$ if this is an information theoretic bound? Can you clarify this claim please? Also, in the context of tensor completion $2^{\rho}>\pi>\sqrt{n}$.

8) What do you mean by "NP-hard to solve to any accuracy"? These words can correspond to different orders of quantifiers, which would result in different claims.

9) L 226 has claim: "Although it is NP-hard, there is substantial structure that enables efficient numerical computation of global minima of (8)." Efficient computation typically stands for "polynomial time", which suggests that you claim P = NP. Clearly, that's not what you claim. I would kindly suggest to reformulate or clarify this claim.

10) In line 249 you claim that your implementation converges to the global optima. However, the experimental results, in the regimes where your theory suggests almost exact recovery (Table 2), have an error in the range 0.01-0.16 (which is much better than other approaches), but is still far from 0. Can you please clarify this?

11) Line 30 says "exponentially more samples than the information-theoretic rate". Can you please clarify what do you mean? Exponential in which variable? The number of samples is always at most linear in the size of the tensor.

**Limitations:**

The paper proposes an approach that is NP-hard in the worst case and this paper does not seem to prove any runtime guarantees. However, the authors show that the problem can be solved in a linear number of calls to an integer-programming oracle (which is still NP-hard). It will be nice to explore special cases in which a polynomial-time algorithm (say, for oracle)  is available.

I believe the paper will also significantly benefit from experiments on real-world data.

**Strengths And Weaknesses:**

Significance: Tensors with non-negative entries frequently appear in practice, hence improved algorithms for this setup are likely to have further applications. This paper proposes a natural idea for the problem, which does not seem to be explored in the prior work. The experimental results presented by the authors show that their heuristic algorithm has better performance on tensors with low non-negative rank compared to some state-of-the-art approaches.

Clarity: The technical proofs seem to be correct and the paper is reasonably well-written. However, there is a number of vague statements, typos, or claims that are potentially overstated. See questions below. Additionally, it will be very helpful for the reader if details of BCG algorithm are included at least in the appendix.

Finally, the paper uses notation that is non-standard in the literature, which makes it a bit harder to read. For instance, $r$ is typically reserved for the rank of the tensor, while this paper uses $k$ for the rank and $r_i$ for the dimensions of the tensor.

---

> ### Author Response · Authors · 2022-08-02
> **Response to multiple questions**
>
> Question 1: We have changed the language to say "achieve substantially closer to the information-theoretic rate" for these past works. Two of the cited methods do not have corresponding results, but we believe an analysis would show this to be the case.
>
> Question 2: We have added a remark to clarify this point. (line 207)
>
> Question 3: We use two oracles in our algorithm. The oracle that gets used the most is implemented as a heuristic, and that is backed up by an oracle that uses an exact MIP solver. The MIP solver is, naturally, worst-case exponential-time since the problem is NP-Hard. Nonetheless our computational experiments demonstrate highly-scalable and data-efficient practical performance. This is because the heuristic oracle is called much more often. The code we include with our paper keeps count of the number of times the MIP-based oracle is called, and it gets called a small number of times when solving problems.
>
> Question 4: We have not focused on a polynomial-time special case, as our contribution focuses on a practical, data-efficient numerical method for the general case. Now, we can show that the separation oracle (and consequently the overall algorithm) can be executed in polynomial-time for the special case of matrix completion. However, in practice it is likely that more specialized methods would still preferred for nonnegative matrix completion. Establishing polynomial-time subclasses of tensor completion via the complexity of the separation problem is an interesting topic, and certainly something we would like to consider in future work.
>
> Question 5: Regarding the difference between $\psi$ and $\widehat{\psi}$ in Prop. 4.1 and Cor 4.3, we have corrected a minor type on the left side of the equation in Prop 4.1 where the ``hat'' was missing. This should now clarify the difference.
>
> Question 7: There was a typo, and little-o should be big-o. We have corrected this typo. Regarding the comment that in tensor completion we have $2^\rho > \pi > \sqrt{n}$, this may not be true in the noisy setting where you could potentially observe several different noisy measurements of each tensor entry:  Tensor completion would give an improvement over simply averaging each observed entry.
>
> Question 9: We have revised the wording to ``...practical numerical computation...''.
>
> Question 10: We report normalized MSE, and the theoretical results are for MSE. Because of how the instances are constructed, each entry in the ``true'' tensor lie between zero and one. Hence, normalized MSE is larger than MSE. Also, the algorithm uses a specified numerical tolerance the determine when to terminate the algorithm, and we note that the number of oracle calls is linear in this numerical tolerance. However, the finite numerical tolerance introduces some amount of error in the reconstruction. Convergence to an optima within a finite numerical tolerance is still considered convergence to global optima within the optimization community.

---

> > ### Comment · Reviewer_4C1A · 2022-08-10
> > **Thank you for the response**
> >
> > Thank you for the response. I am keeping my positive score unchanged for now.
> > I also agree with reviewer 7JBT, that some suggestions that were meant to help improve the presentation of the paper were not sufficiently addressed.

---

### Official Review · Reviewer_7JBT · 2022-07-11

**Rating:** 6
**Confidence:** 3
**Soundness:** 3 good
**Presentation:** 4 excellent
**Contribution:** 3 good

**Summary:**

The authors proposed an approximated algorithm for nonnegative tensor completion that seems to work very efficiently (the claim is in linear time).


**Questions:**

In order to become fully convincing, the paper should report much stronger numerical evidence of scalability (i.e. linear behaviour with n) and efficiency (i.e. ability to work close to the IT threshold).
Moreover, I have to admit to being very surprised that an algorithm that performs a local optimization by flipping single variables one at a time is able to reach the optimal solution in general. The authors should discuss the reasons beyond this unreasonable effectiveness.
I would like to understand whether the apparent very good performance depends on the tensor low-rank (which is not larger than 10 in all numerical experiments shown by the authors). What happens if the tensor rank is increased? Does the algorithm keep working with the same efficiency and scalability?



**Limitations:**

The authors do not really discuss the issue.

**Strengths And Weaknesses:**

The paper is very well written and the authors present their results in a clear way.
The result is definitely original and very important.
My only concern is about the authors' claims based on Figures 3 and 4
The data for the BCG algorithm in the right panels are far from being linear!
Data in Fig. 3 are very noisy and those in Fig. 4 have a clear upwards curvature, suggesting a more than linear growth for larger n values.
Moreover, from the numerical experiments is not clear at all whether the algorithm is matching the IT threshold: data supporting this claim are completely missing or should be presented in a more evident way.

---

> ### Author Response · Authors · 2022-08-02
> **Response on scalability, efficiency, and the algorithm**
>
> Regarding the questions about linear behavior in $n$ and the right side of the figures with computation time, we note that our claim is that the behavior would be linear in \emph{numerical tolerance}. We have revised our manuscript to be more clear that the algorithm is linear in numerical tolerance. We have not characterized the scaling of the algorithm in $n$, but as you point out the numerical results suggest that it is not linear. However, we believe this does not detract from the practical utility of our algorithm because tensor completion is often used in the low $n$ setting.
>
> Regarding numerical evidence of scalability and efficiency, we do note that our experiments scale to tensors with $10^8$ entries, which is comparable to the problem sizes used in research papers to show scalability for matrix completion.
>
> Regarding your surprise that an algorithm that performs a local optimization by flipping single variables one at a time is able to reach the optimal solution in general: We were also surprised by the empirical success of the simple,  heuristic separation oracle, and we have checked our code carefully to confirm. Some of this can be explained by the nature of the BCG algorithm: the separation problem does NOT have to be solved to optimality -- any feasible solution crossing the zero objective value threshold suffices.  Therefore, especially in early iterations, it is relatively easy to find a separating cut.

---

> > ### Comment · Reviewer_7JBT · 2022-08-03
> > **Small rewording does not improve the content on the manuscript**
> >
> > I am somewhat disappointed by the authors' reply.
> > My comments and my requests were really made to help in improving the content of the manuscript and make it more robust (so as to be able to increase my evaluation).
> > However, the authors have decided to make just minimalist rewordings.
> > In my opinion, the new version of the manuscript maintains the weak aspects of the original submission.
> > These aspects could have been improved by running new simulations (which last at most 1.5 hours according to the authors' words) or at least doing a better analysis and presentation, but none of this has been done.

---

### Official Review · Reviewer_skYN · 2022-07-14

**Rating:** 7
**Confidence:** 4
**Soundness:** 4 excellent
**Presentation:** 3 good
**Contribution:** 4 excellent

**Summary:**

The authors introduce an algorithm for nonnegative tensor completion. For that, a new tensor norm is defined which is shown to be a convex surrogate of the tensor rank. To define this norm, the authors first show that the convex hull of the ball of the nonnegative rank-1 tensors whose maximum entry is 1 is the same as the convex hull of all the binary rank-1 tensors. Since the latter is a polytope and the binary rank-1 tensors are the vertices of the polytope, linear integer programming techniques can be used to optimize over it. Therefore, the norm of a nonnegative tensor of any rank is defined using such polytopes. The authors further study the norm by proving its NP-hardness and its stochastic complexity, in terms of Rademacher complexity. Finally, it is shown how integer linear programming can be used for efficient computation of the global minima. The numerical results show that the proposed method can achieve better estimation error and is able to complete tensors with fewer samples.

**Questions:**

It would have been interesting to evaluate the algorithm for hyperparameters other than the ground truth.

**Limitations:**

There is no dedicated "limitations" section. But the authors discuss the limitations, such as the solution being limited to nonnegative tensors, the computational complexity, and the running time.

**Strengths And Weaknesses:**

Strengths:
* The problem of tensor completion is an interesting problem that shows up in different applications from recommender systems to wireless communications. Aside from that, the introduction of a norm that is a convex surrogate of the tensor rank and has nice properties for optimization problems is impactful on its own.
* The authors study the different aspects of the problem and the proposed norm theoretically, prove the claims, and/or refer the reader to the relevant papers.
* A numerical algorithm is proposed to use the introduced norm for the nonnegative tensor completion problem. Although the proposed norm is NP-hard to calculate to arbitrary accuracy, the proposed algorithm is able to calculate the global minima. This is due to the proposed norm being a 0-1 polytope.
* The paper is well-structured and the proofs are well-written. The authors discuss the limitations

Weaknesses:
There are not many weaknesses. The authors could have moved all or some of the proofs to the appendix. This would have opened some space to provide more intuition to the reader, add some visualizations, explain the algorithm in more detail, and possibly add more experiments. For example, it would have been interesting to evaluate the algorithm for hyperparameters other than the ground truth. This is of great practical importance, as we usually do not have access to the true rank or the true maximum value of the tensor.

---

> ### Author Response · Authors · 2022-08-02
> **No response**
>
> We thank the reviewer for their comments.

---

### Meta-Review · Area_Chair_BwNn · 2022-08-27

**Recommendation:** Accept
**Confidence:** Certain

**Metareview:**

All reviewers found the paper clearly interesting, and agree that it makes a valuable and novel contribution in the field of tensor learning, and the consensus to accept the paper is thus without ambiguity.

However, all reviewers, who were all from the start fairly positive about the paper, and who made a number of constructive comments and suggestions that could improve the paper, were quite disappointed by the responses of the authors which seem to suggest that the latter were only prepared to make minimal changes to address the concerns of the reviewers. This explains why the ratings of the paper are not higher...

We obviously understand that it would not have been possible to address some of the concerns of the reviewers during the rebuttal period, but the authors are now strongly encouraged to take into account the comments of the reviewers when preparing the final version of this manuscript. The authors are in particular encouraged to take into account the questions and comments about related work and to the extend possible to include more detailed discussions of the related work and the connections with this work. Also, they are encouraged to clarify the technical parts of the manuscript about which the reviewers asked clarification questions.

**Award:**

No

---

### Decision · Program_Chairs · 2022-09-14

Accept